# Vaccine Hesitancy among European Parents—Psychological and Social Factors Influencing the Decision to Vaccinate against HPV: A Systematic Review and Meta-Analysis

**DOI:** 10.3390/vaccines12020127

**Published:** 2024-01-26

**Authors:** Teodora Achimaș-Cadariu, Andrei Pașca, Nicoleta-Monica Jiboc, Aida Puia, Dan Lucian Dumitrașcu

**Affiliations:** 1Faculty of Medicine, “Iuliu Hațieganu” University of Medicine and Pharmacy, 400012 Cluj-Napoca, Romania; lazea.teodora@elearn.umfcluj.ro (T.A.-C.); aida.puia@umfcluj.ro (A.P.); ddumitrascu@umfcluj.ro (D.L.D.); 2Department of Surgical Oncology, “Prof. Dr. Ion Chiricuță” Institute of Oncology, 400015 Cluj-Napoca, Romania; 3Department of Psychology, Babeş-Bolyai University, 400347 Cluj-Napoca, Romania; nicoleta.jiboc@ubbcluj.ro

**Keywords:** HPV vaccination, hesitancy, Europe, parents, cervical cancer prevention, socio-demographic, psychological factors

## Abstract

Background: Due to low adherence to HPV vaccination programs, the European region struggles with vaccination rates lower than 30% among the targeted population. The present report investigated various socio-demographic and psychological factors influencing European parents’ hesitancy towards vaccinating their children. Methods: As of September 2023, four databases were searched. After applying inclusion criteria, all articles comparing psychological and socio-demographic factors in parents who vaccinate or do not vaccinate their children were included. Results: Twenty-five primary publications met the criteria, totaling 385,460 responders, of which 311,803 want to or have already vaccinated their child, and 73,555 do not wish to do so. Immigrant and employment status, religion, age of parents and the child’s gender seemed to influence their decision to vaccinate their child. Previous experience with vaccines, perceived safety and efficacy and the mother’s previous abnormal CCS results also impacted their decision. The caregivers’ education, gender, civil status, number of children, family history of neoplasia or mother’s CCS attendance did not influence their hesitancy to vaccine. Conclusion: Multiple demographic, social, economic and psychological barriers involved in the parents’ hesitancy to vaccinate their children against the HPV virus were highlighted. Specific at-risk categories that need to be targeted with information, education and vaccination campaigns were identified.

## 1. Introduction

Among female neoplasia cases worldwide, cervical cancer is the pathology currently holding second place concerning its incidence and mortality, despite widely available preventive measures [1,2]. The World Health Organization (WHO) recently proposed the Cervical Cancer Elimination Initiative (also called the 90–70–90 rule), a strategy that aims to fully vaccinate 90% of girls by the age of 15 years old, screen 70% of women with high-performance tests (such as HPV testing or typing) by the age of 35 and again by 45 years old and treat 90% of women with precancerous lesions or invasive cancer by 2030 [3]. Despite this ambitious initiative, from 2021, official cervical cancer screening recommendations were available in only 69% of all countries worldwide, of which 78% were still using cytology as a primary test. Only 35% of them used HPV testing (usually a Polymerase Chain Reaction–PCR test) as a standard, alone or in combination with classic cytology [4].

Meanwhile, the HPV vaccine was approved by the FDA in 2006, initially covering four HPV strains [5]. Currently, Gardasil 9 offers protection against nine strains: 6, 11, 16, 18, 31, 33, 45, 52 and 58 [6]. Despite tremendous efforts to implement vaccination programs worldwide, more than 17 years after its initial approval, only 55% of the WHO Member States introduced the HPV vaccine as a primary prophylaxis measure, and only 33 out of those 107 countries have gender-neutral programs, vaccinating girls and boys [7]. In Europe, 77% of countries have introduced the HPV vaccination [7], the majority with the aid of a National Immunization Program in 2007, shortly after EMA approved the product. Some countries only adhered later, such as Croatia in 2016 [8]. However, different implementation strategies were used, resulting in high heterogeneity regarding vaccine availability. For example, the costs could be partially or fully reimbursed, covered by the National Health Authorities, while other countries offer the vaccine only if fully paid for by the recipients. Moreover, the recommended age for vaccinating children (ranging from 9 to 15 years old) varies significantly between countries regarding the policy of reimbursement and availability through the public health system; some reimburse the vaccine regardless of the ages of the recipients, while only a few offer gender-neutral programs [8]. As of 2022, in the European Region, full vaccination coverage (three doses of HPV vaccine) in females was around 29%, while in males, the percentage was less than half, at around 14%. In total, 38% of females received the first dose, while only 18% of males initiated the vaccination scheme [9]. Most programs targeted young boys and girls, while some countries offered catch-up alternatives; parental consent was required.

This poor coverage of HPV vaccination highlights the discrepancies among European countries regarding vaccination implementation and parents’ (or caretakers’) hesitancy toward initiating or continuing the scheme. Vaccination hesitancy was defined by the World Health Organization (WHO) as the delay in acceptance or refusal despite available access to vaccines [10]. Multiple reasons can influence the decision to vaccinate oneself or one’s child, varying from personal beliefs, insufficient education, low-quality information or a lack of information, safety concerns, a lack of confidence or demanding access to vaccination programs [10,11,12,13]. Therefore, this systematic review and meta-analysis aimed to identify other psychological, social, demographic and economic factors (such as concerns related to the early debut of sexual activity, previous childhood vaccinations, parents’ history of gynecological pathologies, gender of the children, perceived susceptibility and perceived risk, level of education, marital status, income and many more) that could influence European parents’ (or caregivers’) intention to vaccinate their children against HPV.

## 2. Materials and Methods

### 2.1. Research Protocol and Information Sources

A systematic review and meta-analysis were performed to identify psychological and social factors influencing parents’ (or caretakers’) intention to vaccinate their children against HPV. This report followed the Preferred Reporting Items for Systematic Reviews and Meta-Analyses (PRISMA). There was no separate review protocol, and this study was not preregistered. No Ethical Committee approval or informed consent was required since the data were collected from previous publications.

### 2.2. Search Strategy

A search was performed on 13 September 2023 in four databases: PubMed (US National Library of Medicine, Bethesda, MD, USA), EMBASE (Elsevier, Amsterdam, The Netherlands), Scopus (Elsevier), and Web of Science (Philadelphia, PA, USA). A predefined search formula was used to identify relevant studies: ((hpv vaccin*) OR (human papillomavirus vaccin*) OR (human papilloma virus vaccin*)) AND ((knowledge, attitudes, practice) OR (barriers) OR (facilitators) OR (decision) OR (inten*)) AND ((parent*) OR (caregive*) OR (child*)). No additional filters or limits were used.

### 2.3. Eligibility Criteria

Inclusion criteria (formulated to adhere to PICO strategy):Population: Parents and caregivers from European countries (EU-27) and the United Kingdom;Intervention: All studies evaluating psychological, demographic and social factors;Comparison: Vaccinated/intent to vaccinate compared with non-vaccinated/no intent to vaccinate their children;Outcome: Vaccination status/intention to vaccinate;Study type: Self-report quantitative studies, cross-sectional studies and non-experimental studies.

#### Exclusion Criteria

Duplicates, other reviews and meta-analyses, other study designs (qualitative studies, intervention studies correspondence, conference proceedings, books and chapters, validation or development of scale/questionnaire) and studies that assess other topics were excluded. Studies in languages other than English and studies with no available full text were also excluded.

### 2.4. Selection Process

Two reviewers independently screened the studies; if discrepancies were found, a third reviewer solved the issue. No automation tools were used.

### 2.5. Data Collection Process

Two independent reviewers collected data from individual studies, and relevant information was extracted.

### 2.6. Risk of Bias Assessment and Quality Assessment

Funnel plots were used to evaluate the risk of bias. The Newcastle–Ottawa Quality Assessment Scale (NOS), adapted for cross-sectional studies, was used to assess the quality of the included studies [14]. Two reviewers evaluated each study.

### 2.7. Synthesis of Methods and Results

The articles were screened by title and abstract. Articles that met the eligibility criteria (comparison groups, evaluating psychological and social characteristics) were included in the full-text assessment. Eligible articles were included in the systematic review assessing parents (or caretakers) with daughters or sons (regardless of the number of children) who were vaccinated, non-vaccinated or intended to vaccinate. Relevant data were extracted from the included articles: age, gender, educational level, civil status, employment, religion, the immigrant status of parents (or caretakers), number of children, mothers’ CCS attendance and Pap Smear results, family history of any cancer, each child’s gender, previous history of childhood vaccination, perceived efficacy and safety of the HPV vaccine, risks behaviors, HPV knowledge and awareness, costs of the HPV vaccine, perceived consequences, risks and susceptibility decision-making process and barriers toward HPV vaccination.

Combinable data were considered when results were presented either as an already calculated effect of measure (Odds Ratio—OR; Relative Risk—RR; Risk Difference—RD) or numbers were provided regarding the number of events in each group (presence vs. absence of a factor) and the total population number, so that the OR could be calculated. In that instance, OR = (a/c)/b/d), where a = the number of events in the exposed group; b = the number of controls in the exposed group; c = the number of events in the unexposed group; and d = the number of controls in the unexposed group. When other effects of measures were presented (RR or RD), appropriate transformations to OR were performed. If the results were not combinable, a simple systematic review was provided. Odds Ratio (OR) was used as a primary parameter with a Confidence Interval (CI) of 95%. I2 was used to evaluate heterogeneity, and the threshold were set according to Cochrane’s recommendations as follows [15]: 0% to 40%—the heterogeneity might not be important; 30% to 60%—may represent moderate heterogeneity; 50% to 90%—may represent substantial heterogeneity; 75% to 100%—considerable heterogeneity. Fixed or Random Effects models were applied accordingly based on the differences between smaller and larger studies or causes of heterogeneity and bias. As when there is no heterogeneity, a random-effects or fixed-effects model will provide results similar to identical ones, a classical fixed-effects model was applied. Data analysis was performed in Review Manager 5.4 (RevMan). Regarding additional analyses, subgroup analyses were planned if feasible (type of included studies, quality assessment scoring, child’s gender or age, gender of parent), while sensitivity analyses were depicted by excluding one study at a time from the meta-analysis. A good sensitivity was considered to exist when the results remained in the same CI.

## 3. Results

### 3.1. Study Selection

A systematic search was performed in four databases on 13 September 2023, resulting in 9322 articles; after removing the duplicates, 4206 publications were screened by title and abstract. In total, 133 were assessed for full-text, and 25 [16,17,18,19,20,21,22,23,24,25,26,27,28,29,30,31,32,33,34,35,36,37,38,39,40,41] were included in the systematic review and meta-analysis (Figure 1).

Overall, 25 primary publications were included in the systematic review, totaling 385,460 responders (parents or caregivers), of which 311,803 want to or already have vaccinated their child (boy or girl), and 73,555 did not vaccinate or do not intend to vaccinate their child (Table 1). The quality assessment of the included studies can be consulted in Table 2, and a graphical representation can be consulted in Appendix A.

Twenty-eight social, economic, psychological and demographic variables were identified and reported in these publications as influencing the decision to vaccinate. Sixteen factors were presented as combinable data in individual studies and included in subsequent meta-analyses. The rest were evaluated as part of the systematic review and could not be directly compared due to heterogeneity in data-reporting methods. A detailed workflow is available in the Appendix A.

### 3.2. Results of the Systematic Review and Meta-Analysis

#### 3.2.1. Age of Parents

Six studies were included in the meta-analysis of parents’ ages, comparing older (≥40 years old) vs. younger (<40 years old) parents, totaling 106,930 responders, with 83,199 having vaccinated (or intent to vaccinate their) children [16,17,18,19,20,21]. The results revealed an Odds Ratio (OR) of 0.86 in favor of the parents aged less than 40 years old, with a 95% Confidence Interval (CI) between 0.84 and 0.89, at a Z of 9.57 and a *p* < 0.00001. The Fixed Effects (FE) was deployed due to a relatively small heterogeneity (I^2^) of 44% (see Figure 2).

#### 3.2.2. Gender of Parent

Nine studies were included in the meta-analysis of parents’ gender (comparing fathers and mothers), with 18,890 responders, of whom 13,983 had vaccinated (or an intent to vaccinate their) children [17,21,22,23,24,25,26,27,28]. The results revealed an OR of 0.75, with a 95% CI between 0.54 and 1.04, at a Z of 1.71 and a *p* = 0.09; Random Effects (RE) were used for an I^2^ of 89% (See Figure 3).

#### 3.2.3. Parents Educational Level

Fourteen studies were included in the parent’s level of education comparison (ISCED 0–4 vs. 5–8), with 284,944 guardians and 232,063 vaccinated children [17,18,19,20,21,22,24,27,29,30,31,32,33,34]. The results showed an OR of 0.93, with a 95% CI between 0.79 and 1.09, at a Z of 0.94 and a *p* = 0.35, with no statistical significance; Random Effects (RE) were used for an I^2^ of 93% (see Figure 4).

#### 3.2.4. Parents Civil Status

Eight studies were included in the parent’s civil status comparison (single/divorced/widowed vs. married/in a relationship) [17,19,20,22,23,26,29,34]. The results showed an OR of 0.96 with a 95% CI between 0.77 and 1.18, at a Z of 0.4 and a *p* = 0.69, with no statistical significance; Random Effects (RE) were used for an I^2^ of 95% (see Figure 5). 

#### 3.2.5. Immigrant Status

Seven studies [17,20,22,23,24,29,34] were included in this meta-analysis, totaling 173,643 responders, of which 145,458 were vaccinated. Odds Ratio (OR) of 0.61, as well as CIs = 0.38 and 0.98, demonstrated statistical significance at a Z value of 2.05 with a *p*-value = 0.04. Immigrant parents had 39% lower odds of vaccinating their children than native parents in their respective countries (See Figure 6).

#### 3.2.6. Employment Status of Parent

Five studies were included in the meta-analysis, totaling 182,675 responders with 152,710 vaccinated children, comparing unemployed and employed parents [19,20,26,29,34]. The results exhibit an OR = 0.45, with a 95%CI between 0.28–0.74, at a Z = 3.18, *p* = 0.001, and I^2^ = 99%. Random Effects were used due to the high heterogeneity. Unemployed parents had 55% lower odds of vaccinating their children than employed parents (See Figure 7).

#### 3.2.7. Religion of Parent (Non-Religious vs. Religious)

Seven studies were included in the meta-analysis comparing any religious faith with no faith [16,22,23,26,28,32,35]. The results demonstrate an OR = 0.75, with a 95% CI between 0.64 and 0.89, at a Z = 3.39, with a *p* = 0.0007 and I^2^ = 47%; FE were used as a model. Children in religious households had 25% lower odds of getting vaccinated (See Figure 8).

#### 3.2.8. Number of Children

Five studies were included in the comparison regarding the number of children in a household (>two children vs. ≤2) [16,17,20,26,28]. No statistical significance was highlighted, showing an OR of 0.97, with a 95% CI between 0.94 and 1, at a Z = 1.77, with *p* = 0.08; FE were used at a trivial heterogeneity of I^2^ = 0% (See Figure 9).

#### 3.2.9. Mothers History of Cervical Cancer Screening (CCS)

Seven studies were included in the meta-analytical comparison between mothers with attendance and non-attendance at CCS and comprised 204,073 responders, of whom 160,699 had vaccinated children [19,20,23,30,32,33,36]. No differences were shown between groups, with OR = 0.83, with a 95% CI between 0.45 and1.52, at a Z value of 0.61, *p* = 0.54 and a 98% heterogeneity. RE were used due to higher heterogeneity (See Figure 10).

#### 3.2.10. Mother Pap-Smear History

Seven studies summing up 189,039 mothers, of whom 149,766 had vaccinated their children, were included in the meta-analysis comparing mothers’ Pap smear histories (normal vs. abnormal results) [19,20,23,30,33,36,37]. Mothers who had a history of abnormal Pap smear results were 46% more likely to vaccinate their child, evidencing an OR = 0.54, with a 95% CI between 0.36 and 0.79, at a Z = 3.16, with *p* = 0.002 and I^2^ = 90%, using RE (See Figure 11).

#### 3.2.11. Family History of Cancer 

Four studies were included in the family history of cancer comparison [23,26,28,32]. No association was found between a positive family history and the intention to vaccinate children, showing an OR of 0.89, with a 95% CI between 0.70 and 1.13, at a Z = 0.94, *p* = 0.35 and low I^2^ = 22% (See Figure 12).

#### 3.2.12. Child Gender (Boys vs. Girls)

Five studies comprising 16,856 responders with 12,508 vaccinated children were included in the comparison regarding the child’s gender [17,18,21,23,27]. Boys were 68% less likely to be vaccinated by their parents than girls, with an OR = 0.32, a 95% CI = 0.13–0.76, a Z value of 2.60, *p* = 0.009 and a high I^2^ = 98% (See Figure 13).

#### 3.2.13. Previous History of Childhood Vaccination 

Eight studies [20,21,22,23,25,30,37,38] were included in the meta-analysis containing 96,211 parents and 74,442 vaccinated children. Pooled analysis showed an OR of 0.22, CI = 0.12–0.4, Z = 4.94 and *p* < 0.00001. Parents who previously refused a vaccine for their child or whose child is not fully vaccinated (with the mandatory schemes) have 88% fewer chances to vaccinate their children against HPV (See Figure 14).

#### 3.2.14. Perceived Efficacy of HPV Vaccination

This meta-analysis included six studies [17,21,25,26,32,37] with 18,554 responders, of whom 13,481 had vaccinated their children. The results demonstrate an OR of 0.24, CI = 0.16–0.37, Z = 6.44 and *p* < 0.00001. Parents who believed the vaccine to be efficient had 76% greater odds of vaccinating their children (See Figure 15).

#### 3.2.15. Perceived HPV Vaccine Safety

This meta-analysis included six studies [17,25,26,32,37,38] and 17,546 parents, of which 12,832 had vaccinated/intended to vaccinate their children. The results revealed an OR of 0.3, CI = 0.18–0.5, Z = 4.69 and *p* < 0.00001 (See Figure 16). Responders who thought the vaccine was unsafe had 70% lower odds of vaccinating their child. The main concerns were fear of side effects (long or short term) [16,17,19,23,25,26,28,29,30,32,33,35,37,38,39], too new or a lack of research [19,23,25,30,33,37], fear of needles/injection or pain [35] and a previous bad reaction to vaccination or regret [38].

#### 3.2.16. HPV Vaccination Would Encourage Sexual Activity

Seven studies [17,22,25,26,32,37,39] totaled 18,326 responders, of whom 13,132 had vaccinated their children and were included in the meta-analysis. The outcome showed an OR of 0.56, CI = 0.39–0.79, Z = 3.31 and *p* = 0.0009 (See Figure 17). Caregivers who believed that the vaccination would increase the at-risk behavior of their child had lower chances of vaccinating their child. Also, believing that their child had a girlfriend or boyfriend or were sexually active positively influenced the intention to vaccinate, while believing that vaccination would decrease condom usage [21] negatively impacted the intention to vaccinate.

#### 3.2.17. Knowledge Related to HPV 

Knowing that the HPV vaccine prevents CC, knowledge related to HPV infection and a positive attitude towards vaccination could influence a parent’s intention to vaccinate their children [21,24,26,27,28,37,38]. In contrast, a lack of or insufficient knowledge about HPV (or the HPV vaccine) could discourage parents from vaccinating their children; other detrimental factors identified were valuing personal knowledge more than experts’ opinions and having a positive attitude toward social media stories or Internet-based information about the vaccine [22,25,28,29,30,32,39]. Five studies found no association between HPV knowledge (regarded as an STI), HPV vaccine knowledge or knowledge related to cervical cancer screening [19,22,26,37,40] and parents’ intention to vaccinate one child.

#### 3.2.18. Recommendation

Receiving the HPV vaccine recommendation from a General Practitioner (GP), a Healthcare Professional (HCP), or a gynecologist [26,28,30,33,37,41] seemed to positively influence the parent’s intention to vaccinate their children [36], as opposed to obtaining the recommendation online. In contrast, parents who were advised against HPV vaccination by relatives, friends or HCPs and those who did not receive any recommendations had lower odds of vaccinating their children [22,28,30,33].

#### 3.2.19. Awareness of HPV

Prior awareness of HPV or receiving education from a GP [17,25,26,41] could positively affect the parents’ intention to vaccinate their children, while being unaware of the HPV vaccine’s existence or purpose negatively influenced their intention to vaccinate [28,33,41].

#### 3.2.20. Need for Information

Receiving information from a GP and Health Authorities or a HCP [24,41] correlated with vaccination status. Caregivers with a higher intention to vaccinate their children [18,25,28,32,37] usually displayed the need for more information to decide, received positive information materials from social media and displayed a higher rate of information searches. One study found no association between vaccination intention and the need for more information regarding HPV, cervical cancer (CC) or HPV vaccination [23]. Searching for information due to mistrust or not receiving guidance from a GP; limited, biased, or unclear information provided by the Government regarding HPV vaccination; a lack of information and feeling the need for more information; or receiving damaging information about vaccination from the media adversely influenced the intention to vaccinate [18,24,28,32,41].

#### 3.2.21. Decision-Making Process and Involvement

Involving the other parent in the decision-making process or thinking that the other parent would want to vaccinate their children positively influenced the decision to vaccinate [25,41]. At the same time, one study found no association between the inclusion of the other parent of the child in the decision-making process and the intention to vaccinate [22]. Believing that the children’s opinion is essential had a positive association with the vaccination intent [23,33]. In contrast, involving their children in the decision-making process or waiting for the children to ask for vaccination negatively influenced parents’ decisions [19,41]. Caregivers believing that their child was incapable of making their own decision regarding vaccination did not seem to influence their decision to vaccinate [32].

#### 3.2.22. Communication about Sexual Life and Vaccination

Parents with difficulties in talking openly with their children, a doctor or any HCP regarding sexual health did not seem to have lesser chances of vaccinating their children [37]. Interestingly, parents who did not address the sexual issues or the HPV vaccine seemed more unlikely to want to vaccinate their children [16,19,37].

#### 3.2.23. HPV Vaccine Cost

The cost of the HPV vaccine correlated with the parents’ decision to vaccinate their children. Having to pay for the vaccine was associated with a lower intention to vaccinate [17,19,28]; reimbursement was mentioned as a solution [28], while one study found no such association [18]. Willingness was higher among girls’ parents than those of boys, even if they had to pay for the vaccine [17].

#### 3.2.24. Perceived Severity and Consequences

Caregivers believing that vaccination assists in preventing a severe disease or health consequences had higher chances of vaccinating their children [19,25]. In contrast, those who believed that HPV infection is not severe were less likely to vaccinate [32]. Not being worried about HPV infection or its consequences did not seem to affect vaccination rates [22,26,32].

#### 3.2.25. Perceived Susceptibility

Feeling that the child would be at risk of contracting STDs or that the child could be infected with HPV had a positive effect on parents’ vaccination intent [16,21,25]; on the contrary, low perceived susceptibility of becoming infected or developing CC [32,39] influenced the intention unfavorably, and one study found no such association [22].

#### 3.2.26. Fear of HPV-Related Disease and Health Concerns

Thinking that vaccination is beneficial or that it can empower parents to protect their children’s health and fear of HPV-related diseases encouraged parents to vaccinate their children [22,28,30,33,38].

#### 3.2.27. Mandatory Vaccination

The non-mandatory character of the HPV vaccine was a paramount factor affecting the parents’ decision to vaccinate, shaping safety concerns and leading to many parents deeming it unnecessary [28,35,37]. In contrast, some parents thought everyone should choose whether or not to vaccinate their children [37].

#### 3.2.28. Trust in the Healthcare System

Parents’ trust in pharmaceutical companies, the government, physicians, pharmacists, researchers and the mainstream media to stop vaccination if serious consequences appeared [23,28,37,38] favorably influenced their intention to vaccinate. A lack of or low trust in the government and the limited information provided, a high number of necessary vaccines, concerns related to fierce side effects and beliefs that pharmaceutical companies influence the government were among the reasons invoked by parents for vaccine refusal [23,32].

#### 3.2.29. Perceived Risks of the Vaccine

Perceiving more risks than benefits regarding the HPV vaccination lowered the parents’ intention to vaccinate their children [22,33,35]. 

#### 3.2.30. Suitable Age for Vaccination

Perceiving their child to be too young to be vaccinated against HPV negatively influenced the decision to vaccinate [19,30,32,35,37], while two studies found no such association [21,33]. One study found parents’ preference for vaccinating sons younger than 12 [24].

#### 3.2.31. Barriers Related to the HPV Vaccination

The main barriers encountered by parents were the need for many doses (2 or 3), the school not offering the HPV vaccine or their daughter not being enrolled at a school [17,35]. Three studies found no association between locations, the number of necessary doses or the complicated steps involved in getting vaccinated [22,32,37]. Interestingly, one study found parents proposing that the GP prepare the vaccine in the office for same-day vaccination [37].

#### 3.2.32. Gender Neutral Vaccination

Most parents would agree to vaccinate their children if the vaccine was gender-neutral [23,25,32].

#### 3.2.33. Income

Higher-income status was associated with vaccination intent in some studies; however, regarding immigrant parents’ income, lower income was associated with the intention to vaccinate [20,31,34,41]. Lower paternal income and lower income for non-immigrant parents were adversely associated with vaccination intention [20,34].

### 3.3. Additional Analyses

Sensitivity analyses were conducted for each meta-analysis, excluding one study. For the majority of analyses, robust sensitivity was highlighted. Interestingly, for three comparisons, an excluded study produced an overall OR outside the initial 95%, making the sensitivity of those comparisons inadequate. Therefore, the meta-analyses (the number of children, the mother’s Pap smear history and parents’ age) were performed again without the respective problematic studies. The results stayed the same. Interestingly, for one comparison (mother’s Pap smear history), the heterogeneity dropped significantly from 90% to 43%, although the overall results were unaffected. From the planned subgroup analyses, only one was possible from the planned subgroup analyses based on the quality assessment scoring on the NOS evaluation. Since all the included studies were cross-sectional, the subgroup analyses based on the publication type were impossible. Most articles interviewed mothers or generically described “parents” or caregivers; therefore, comparisons between fathers and mothers were deemed impossible. Regarding children’s gender, not enough data were given to compare males vs. females, with most studies including just females or both genders. At the same time, only one publication included just male children, making the subgroup analysis impossible (due to only one study being included in this category). The heterogeneity presented in the child’s age category significantly affected the possibility of splitting the publication into different age categories. Finally, the NOS scoring data allowed for the split into mediocre and high-quality studies for six comparisons (parents’ education level, civil status, immigration status, employment, the mother’s history of Pap smear results and previous experience with CCS). All other comparisons only included mediocre publications or had just one study with high NOS scoring, making the split into subgroups impossible. The statistically significant effect was only carried over in the high-quality subgroups for the parent’s employment status and the mother’s previous Pap smear results. In contrast, results were insignificant in the low-quality subgroup. For the mother’s previous Pap smear results, the subgroup analysis was performed both with and without Dib et al.’s [37] study, since this publication yielded problematic results in the sensitivity analysis. The results remained the same. Intriguingly, for the parent’s immigration status comparison, the effect only occurred in the low-quality subgroup, while results seemed to be insignificant in the higher-quality publications subgroup. All additional analyses can be consulted in Appendix A. The original RevMan workflow file can also be consulted in Appendix A.

## 4. Discussion

To the best of our knowledge, this is the most comprehensive systematic review and meta-analysis investigating the psychological, social, demographic and economic factors influencing European parents’ decision to vaccinate their children against HPV. The report efficiently identifies social determinants for at-risk categories of parents who would not vaccinate their children: aged more than 40 years old, with immigrant status, unemployment, and religiousness all compounding factors [16,18,19,20,21,22,23,24,26,28,32,34,35,41]. Some studies highlighted that marital status could contribute to hesitancy, while the current meta-analysis showed that civil status had no influence [17,19,20,22,23,26,34,41,42]. Immigrant status was correlated with lower knowledge of HPV and lower intention to vaccinate children against HPV [43,44]. These results emphasise a need for more information with simultaneous social and economic support for at-risk categories. Policies must also consider the discrepancies between European countries regarding the costs of the vaccine, health insurance coverage and out-of-pocket expenses in various regions for these specific at-risk categories to whom general policies might not apply [17,19,28,45,46,47].

A previous history of abnormal Pap smear was correlated with higher HPV vaccine acceptance, while only attending CCS was not associated, suggesting that CCS could create a false belief that screening could be the only method required and would enough protection against CC [19,20,23,30,32,33,36,37]. These results potentially show a need for more knowledge, information and recommendations by HCPs to make an informed decision [18,21,24,25,26,28,30,33,37,38,41,42,48,49,50].

The gender of parents did not influence the decision to vaccinate their child, while the gender of the child showed a higher chance for girls to be vaccinated compared to boys [17,18,21,22,23,24,25,26,27,28]. The systematic review showed a preference for gender-neutral vaccination [23,25,32], possibly highlighting more trust in the efficacy and safety of HPV vaccination while not putting one’s child at risk; in agreement with this, one study highlighted parents of boys advocating for gender-neutral vaccination [51].

The main reasons for refusing the HPV vaccines were fear of side effects and the novelty of the product [17,19,23,25,28,30,32,33,35,37,38,42,45,46,47,50,51,52,53,54,55,56]. In contrast, a previous history of childhood vaccination and the perceived efficacy and safety of the HPV vaccine positively influenced the intention to vaccinate one’s child [21,22,23,25,26,30,32,37,38,47,57], contouring a profile for parents who trust and believe that vaccines are crucial for their child’s health.

More reasons for refusing vaccination were the belief that the vaccine would encourage at-risk sexual activities [17,22,25,26,32,37,39,56], believing that one’s child is too young to receive the HPV vaccine while showing a preference for older-age initiation of vaccination [19,30,32,35,37,48]. Other factors were parents’ difficulty discussing with their children the purpose of the HPV vaccine or about sexual life and sexual activity, with parents invoking that these types of discussions could prompt their children to be more curious and encourage risky sexual behavior.

Perceived high susceptibility to contracting the HPV infection, believing one’s child is at risk for severe consequences and perceiving more benefits of HPV vaccination seemed to encourage parents to vaccinate. In contrast, low perceived susceptibility, unproven benefits and more barriers to receiving the HPV vaccine could have the opposite effect [16,19,21,22,25,26,32,33,35,39,42,46,56,57]. In contrast, fear or anxiety regarding a possible HPV infection and anticipating consequences could act as determinants for HPV vaccination.

Interestingly, many of the aforementioned factors influencing vaccine hesitancy were superimposable to those present in various regions around the world such as Africa and Asia [58,59,60,61], although presented data highlighted some worrisome statistics regarding the availability of such products. Moreover, extensive data were published regarding the cost-effectiveness of implementing vaccination programs in these regions [62].

Logistical steps related to the availability of the vaccine, convenience, and the repeated character of the HPV vaccine doses, as well as the lack of school-based vaccination [17,35], have been shown to impede parents from vaccinating their children [45,53] emphasising the efforts necessary to make HPV vaccination readily available with uncomplicated procedures.

Mistrust in HCPs and the health system also discouraged parents from opting for the vaccine. At the same time, the dissimilarities between policies create a lack of sustainable support, all contributing to making parents more hesitant [23,32,51,54]. The non-mandatory character of HPV vaccination [28,35,37] adversely impacted a parent’s decision to vaccinate.

### 4.1. Limits

This is the most exhaustive investigation linking miscellaneous psychological, social and demographic factors to vaccine hesitancy among European parents. Although a substantial number of studies was initially screened, and the analyses were based on a pool of more than 380,000 responders, some constraints must be acknowledged. Concerning the methodology, the PICO inclusion criteria could be considered a limitation of the present investigation. By confining the publication to the European region, a significant number of studies were excluded from the present analyses (58 publications), potentially spoiling the results. Although further investigations could be undertaken, and exciting comparisons could arise from such calculations, the purpose of the present study was not to investigate other regions. Language criteria (English language only) further restricted the number of included studies. Eleven publications were excluded solely based on this criteria. Given that English is the primary language in only one country included in this study (the United Kingdom, which is also not a part of the European Union anymore), this could be considered a setback contributing to publication bias and heterogeneity. It is also paramount to mention the search formula used to scan all the databases. Although synonyms were employed where possible, and MeSH was used for indexing correct terms, some restrictions can be highlighted, such as the lack of terms concerning fathers. This was mitigated by including words like caregivers, which comprise both genders, but results explicitly show fewer studies investigating fathers alone as responders. Even so, a considerable number of studies were screened from the databases. Database choice can also affect the number and types of articles included. For the present investigation, four noteworthy databases were searched, and although a relatively large number of studies was uncovered (more than 9300 publications), half were duplicates. References included in the most paramount studies in this field were carefully screened, but unfortunately, no other publications were identified through other sources. This issue can also contribute to the nuanced publication bias highlighted in the analyses.

The quality assessment of the included studies was accomplished using a modified NOS scale and highlighted modest-to-good-quality reports, which can be considered another detriment. Cut-off values for the NOS scoring allowed for further subgroup analyses resembling these categories and helped to distinguish results based on the quality of the included publications. This is virtually impossible to mitigate and instead shows the mediocre quality of the studies in this field while highlighting the need for more high-quality research investigating HPV vaccine hesitancy. The lack of RCTs scrutinizing the current matter means there were no available high-quality studies to compare.

Although no considerable publication bias was emphasized (except for the slight warping of the Funnel Plots that can be attributed to the smaller studies exaggerating the effect), some of the heterogeneity was lessened by excluding studies with the help of sensitivity analyses and performing subgroup analyses. Nonetheless, significant heterogeneity remained even after the investigations above and due to data heterogeneity, and no further explorations were possible. We tried explaining the heterogeneity of the included studies (I^2^ = 0–99%), and perhaps it could be attributed to the population heterogeneity; the geographic discrepancies regarding the HPV vaccination status, incidence, prevalence and mortality; and the types of included studies.

### 4.2. Future Directions

Noteworthy discrepancies could be indicated throughout European countries regarding the implementation, reimbursement, availability, recommendations and addressability of HPV vaccination programs. Despite tremendous efforts to increase addressability, the European vaccination programs still need help with coverage, which was as low as 30% for females and 14% for males. In comparison, only 77% of European countries have implemented them. These worrisome statistics highlight the need for unified policies at the regional, national and European levels. Such strategies could improve the discrepancies between countries while mitigating the low coverage in precise at-risk categories, such as low-income households or immigrants, through shared funding or universal coverage, regardless of the insurance status of the recipients. The present report identified these distinctive socio-demographic characteristics in households that require intensive awareness and education programs, perhaps regulated at the central level: older caregivers, religious households, unemployed parents and immigrants. More aggressive campaigns must reach these communities, empower these parents to protect their children’s health and safety and improve vaccination coverage. 

Detrimental factors, such as complicated procedures, a lack of school-based vaccinations and a lack of invitation-based programs can also be mitigated by improving the delivery system for the vaccine. The present report also highlighted the intricate relationship between the mandatory status of the vaccination and parents’ beliefs, potentially implying a need to implement HPV vaccination as a part of mandatory schemes in a gender-neutral program.

In order to facilitate information and knowledge dissemination among parents, GPs could have a pivotal advocacy role in vaccination campaigns; they are also in direct contact with other at-risk categories, monitoring parents with previous adverse histories regarding vaccinations while being involved in CCS programs. This can be crucial as various interventions can be delivered to mothers that address secondary preventive measures (CCS), highlighting the complementarity of both strategies (primary and secondary prophylaxis). HCPs are also valuable in delivering further recommendations regarding vaccination, as many psychological factors were deemed detrimental to the parent’s intention to vaccinate their children, such as the perceived safety and efficacy and personal misbeliefs regarding risky behaviors after vaccination. 

To conclude, unified, simplified, accessible and fair measures must be implemented broadly. At the same time, more aggressive information and education campaigns must be conducted by trusted HCPs, especially in distinct at-risk categories, to improve HPV vaccination addressability and decrease hesitancy among parents. Specific at-risk categories that need to be targeted with information, education and vaccination campaigns were identified.

## 5. Conclusions

This systematic review and meta-analysis identified crucial determinants that could negatively influence parents’ intention to vaccinate their children: parents older than 40, immigrant status, being unemployed, religious beliefs, previous childhood vaccination refusal, perceiving the vaccine to be unsafe, claiming fear of side effects and believing that children vaccinated against HPV would be encouraged to perform at-risk sexual activities or would result in an early debut. A history of abnormal Pap smear results in mothers and perceiving the HPV vaccine as being effective in preventing diseases would encourage parents to vaccinate their children.

## Figures and Tables

**Figure 1 vaccines-12-00127-f001:**
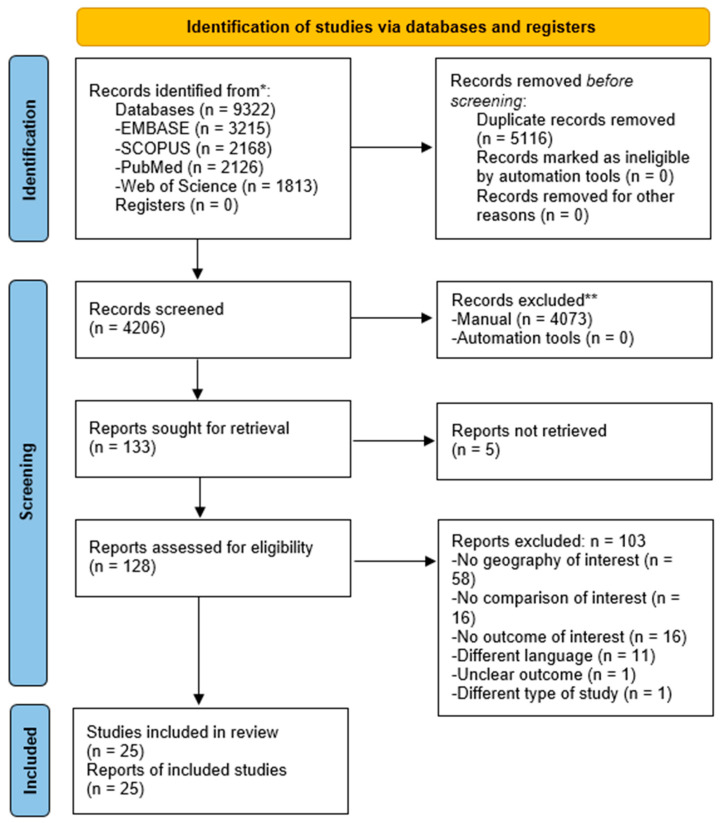
The PRISMA flow chart shows the process involved in searching and filtering the primary publications. * Number of records identified from each database or register searched. ** If automation tools were used, indicate how many records were excluded by a human and how many were excluded by automation tools.

**Figure 2 vaccines-12-00127-f002:**
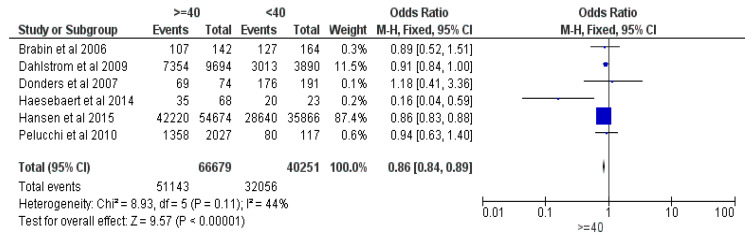
Forrest plot for the age of parent comparison [16,17,18,19,20,21].

**Figure 3 vaccines-12-00127-f003:**
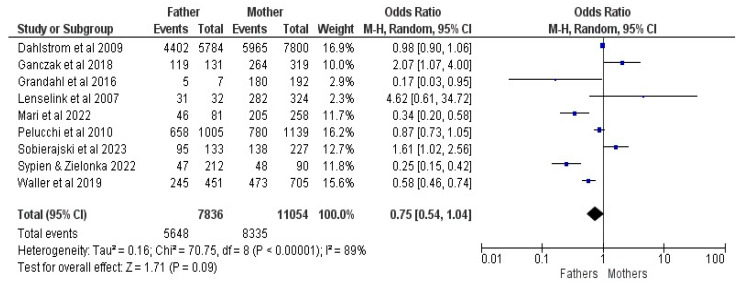
Forrest plot for the gender of parent comparison [17,21,22,23,24,25,26,27,28].

**Figure 4 vaccines-12-00127-f004:**
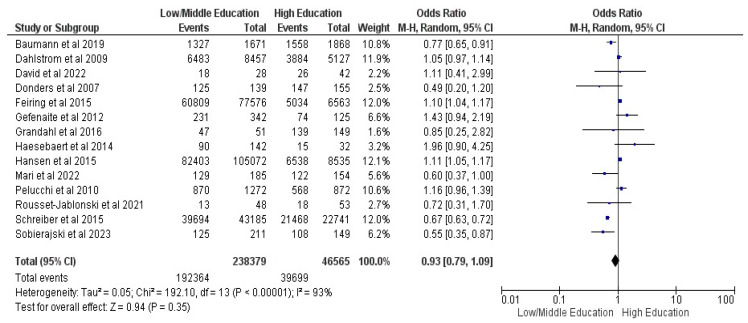
Forrest plot for the parent’s educational level comparison [17,18,19,20,21,22,24,27,29,30,31,32,33,34].

**Figure 5 vaccines-12-00127-f005:**
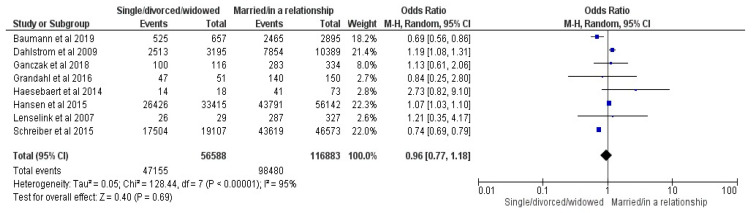
Forrest plot for the parent’s civil status comparison [17,19,20,22,23,26,29,34].

**Figure 6 vaccines-12-00127-f006:**
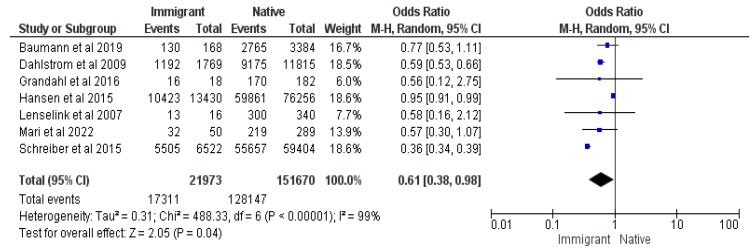
Forrest plot for the immigrant status comparison [17,20,22,23,24,29,34].

**Figure 7 vaccines-12-00127-f007:**
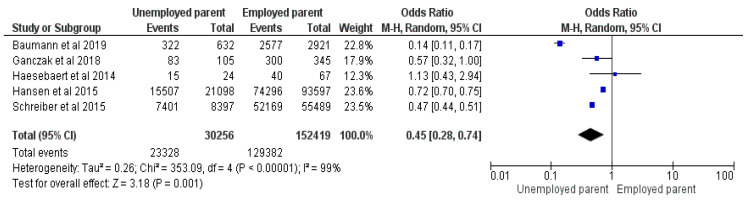
Forrest plot for parent employment status comparison [19,20,26,29,34].

**Figure 8 vaccines-12-00127-f008:**
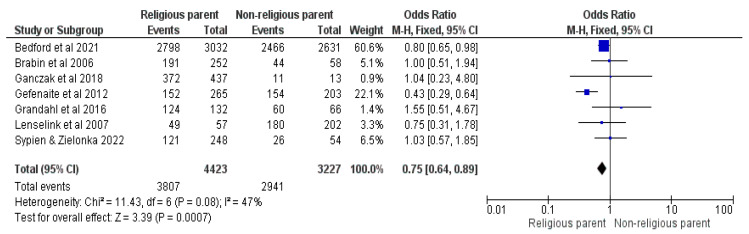
Forrest plot for the comparison of the religious statuses of the responders [16,22,23,26,28,32,35].

**Figure 9 vaccines-12-00127-f009:**
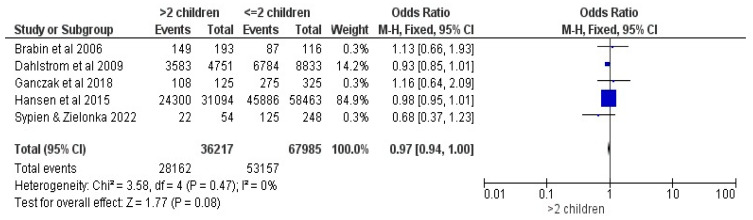
Forrest plot for the comparison of the number of children [16,17,20,26,28].

**Figure 10 vaccines-12-00127-f010:**
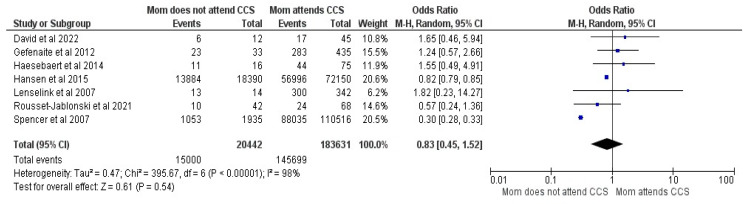
Forrest plot for the mother’s history of CCS [19,20,23,30,32,33,36].

**Figure 11 vaccines-12-00127-f011:**
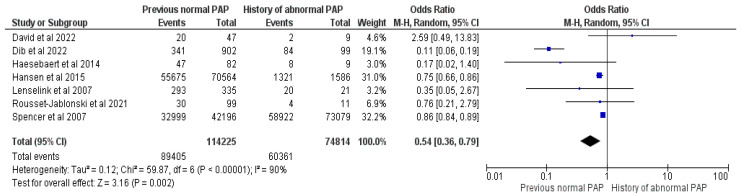
Forrest plot for the comparison of the mother’s Pap smear history [19,20,23,30,33,36,37].

**Figure 12 vaccines-12-00127-f012:**
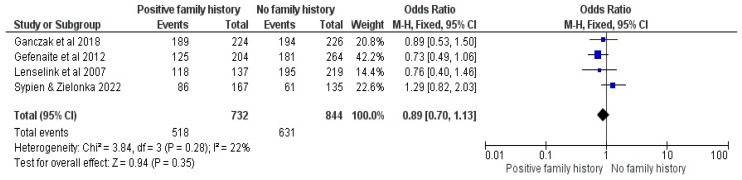
Forrest plot for the comparison of the family history of cancer [23,26,28,32].

**Figure 13 vaccines-12-00127-f013:**
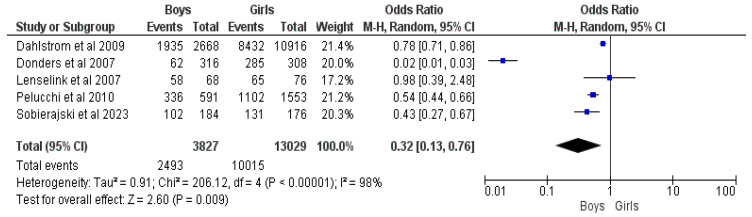
Forrest plot for the comparison of the child’s gender [17,18,21,23,27].

**Figure 14 vaccines-12-00127-f014:**
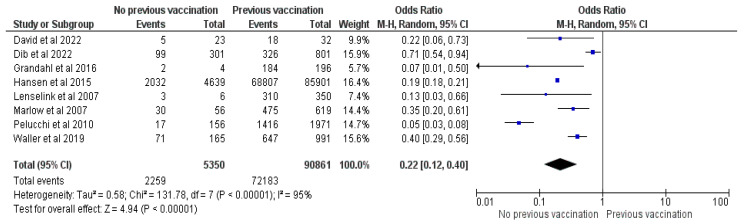
Forrest plot for the comparison of previous vaccination of child [20,21,22,23,25,30,37,38].

**Figure 15 vaccines-12-00127-f015:**
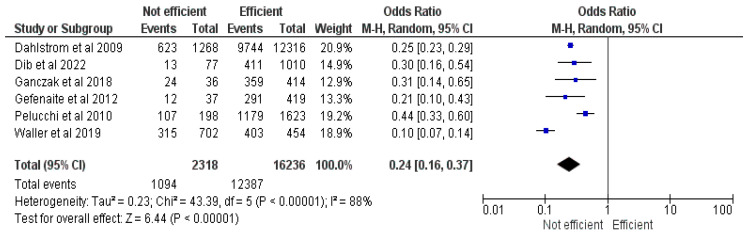
Forrest plot for the comparison of perceived efficacy [17,21,25,26,32,37].

**Figure 16 vaccines-12-00127-f016:**
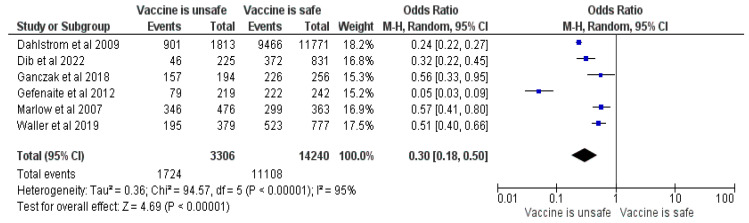
Forrest plot for the comparison of perceived safety [17,25,26,32,37,38].

**Figure 17 vaccines-12-00127-f017:**
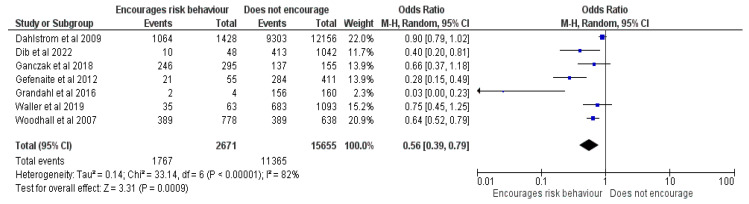
Forrest plot for the comparison of the perceived increased risk behavior after vaccination [17,22,25,26,32,37,39].

**Table 1 vaccines-12-00127-t001:** Characteristics of the included studies.

Study	Country	Respondents	Parent’s Age (in years)	Gender of Children	Vaccination Status	Children’s Age (in years)	Recruitment Site	Data Collection Period	Sample Size	Number of Vaccinated Children	Number of Non-Vaccinated Children
Baumann et al. (2019) [29]	Denmark	Mothers	<20–>40 (at parturition)	Female	Vaccinated vs. non-vaccinated	13	Danish Civil Registration System (CRS)	2015–2016	3558	2899	659
Bedford et al. (2021) [35]	United Kingdom	Parents	Not specified	Female	Initiated vs. non-initiated	14	Millennium Cohort Study (MCS)	2014–2016	5654	5265	399
Brabin et al. (2006) [16]	United Kingdom	Parents	≤30–>45	Not specified	Consenting vs. non-consenting	11–12	Manchester City Council’s Education Department	2005	317	239	78
Dahlström et al. (2009) [17]	Sweden	Mothers and fathers	<41–>45	Female and male	Intention vs. non-intention	12–15	Swedish Population Register	2007	13,946	10,537	3303
David et al. (2022) [30]	France	Parents	32–58	Female	Vaccinated vs. non-vaccinated	9–19	Hospital Femme-Mère Enfant and Necker-Enfants-Malades	2020–2021	71	28	43
Dib et al. (2022) [37]	France	Mothers	43.5 (mean)	Female	Vaccinated vs. non-vaccinated	11–14	Not specified	Not specified	1102	425	677
Domínguez-Riscart et al. (2023) [40]	Spain	Caregivers (mostly mothers)	44.1 (mean)	Trans-gender	Intention vs. non-intention	9–16	Hospital Puerta del Mar and Universitario Reina Sofia	2022	65	14	51
Donders et al. (2007) [18]	Belgium	Mothers	35.8 (mean)	Female and male	Intention vs. non-intention	Not specified	Hospital Hart	2007	309	285	23
Feiring et al. (2015) [31]	Norway	Mothers and fathers	≤25–>35 (at parturition)	Female	Initiated vs. non-initiated	Not specified	The Norwegian Central Population Registry	2013	84,139	65,843	18,296
Ganczak et al. (2018) [26]	Poland	Mothers and fathers	29–67	Not specified	Intention vs. non-intention	13–16	High School	2013–2014	450	383	67
Gefenaite et al. (2012) [32]	Netherlands	Parents	35–55	Female	Vaccinated vs. non-vaccinated	13	Institute for Public Health and the Environment	2009	469	307	162
Grandahl et al. (2016) [22]	Sweden	Parents	43.2 (mean)	Children	Consenting vs. declining	11–12	Swedish National Population Register	Not specified	200	186	14
Haesebaert et al. (2014) [19]	France	Mothers	18–65	Female	Favorable vs. uncertain/opposed	14–18	General Practitioner	2008	91	55	36
Hansen et al. (2015) [20]	Norway	Mothers and fathers	<35–≥50	Female	Initiated vs. non-initiated	12–13	Norwegian Immunisation Registry (SYSVAK)	2013	90,540	70,870	19,670
Lenselink (2007) [23]	Netherlands	Mothers and fathers	42.2 (mean)	Female and male	Intention vs. non-intention	10–12	School	Not specified	356	313	32
Mari et al. (2022) [24]	Italy	Parents	44.5 (mean)	Male	Intention vs. non-intention	6–18	Buzzi Hospital	2021–2022	339	251	88
Marlow et al. (2007) [38]	United Kingdom	Mothers	41.1 (mean)	Female	Consenting vs. non-consenting	8–14	School	2006	684	513	171
Pelucchi et al. (2010) [21]	Italy	Mothers and fathers	<40–≥50	Female and male	Consenting vs. non-consenting	10- 13	School	2008	2144	1438	706
Rousset-Jablonski (2021) [33]	France	Accompanying adults (parents)	32–58	Female	Vaccinated vs. non-vaccinated	9–17	Pediatric or pediatric and adult Cystic Fibrosis centers	2018–2019	104	34	76
Schreiber et al. (2015) [34]	Denmark	Mothers	Not specified	Female	Initiated vs. non-initiated	Not specified	Danish Civil Registration System	Not specified	65,926	61,162	4764
Sobierajski et al. (2023) [27]	Poland	Parents	<30–>49	Children	Vaccinated/intention vs. non-vaccinated/non-intention	9–15	Central Statistical Office’s data	2022	360	233	127
Spencer et al. (2013) [36]	United Kingdom	Mothers	25–65	Female	Initiation vs. non-initiation	12–15	National Health Authority Information System	Not specified	112,451	89,088	23,363
Sypien & Zielonka (2022) [28]	Poland	Parents	≤34–>34	Female and male	Intention vs. non-intention	Not specified	University Children’s Hospital	2021	302	95	207
Waller et al. (2019) [25]	United Kingdom	Parents	40.5 (mean)	Female and male	Decided vs. non-decided/undecided	9–12	School	2019	1156	718	438
Woodhall et al. (2007) [39]	Finland	Parents	≤40–≥51	Female and male	Accepting vs. declining	15 (mean)	Tampere city	2005	727	622	105

**Table 2 vaccines-12-00127-t002:** NOS scoring for the included studies. The Newcastle–Ottawa Scale (NOS), adapted for cross-sectional studies, was used.

Study	Selection (Out of 5 *)	Comparability (Out of 2 *)	Outcome (Out of 3 *)	TOTAL (Out of 10 *)
Baumann et al. (2019) [29]	3	2	2	7
Bedford et al. (2021) [35]	3	2	2	7
Brabin et al. (2006) [16]	3	2	2	7
Dahlström et al. (2009) [17]	3	2	2	7
David et al. (2022) [30]	1	2	2	5
Dib et al. (2022) [37]	2	2	2	6
Domínguez-Riscart et al. (2023) [40]	1	2	2	5
Donders et al. (2007) [18]	1	2	2	5
Feiring et al. (2015) [31]	4	2	3	9
Ganczak et al. (2018) [26]	1	2	2	5
Gefenaite et al. (2012) [32]	3	2	3	8
Grandahl et al. (2016) [22]	1	2	2	5
Haesebaert et al. (2014) [19]	1	2	2	5
Hansen et al. (2015) [20]	3	2	3	8
Lenselink (2007) [23]	1	2	2	5
Mari et al. (2022) [24]	1	2	2	5
Marlow et al. (2007) [38]	1	2	2	5
Pelucchi et al. (2010) [21]	1	2	2	5
Rousset-Jablonski (2021) [33]	1	2	2	5
Schreiber et al. (2015) [34]	3	2	3	8
Sobierajski et al. (2023) [27]	3	2	2	7
Spencer et al. (2013) [36]	3	2	3	8
Sypien & Zielonka (2022) [28]	1	2	2	5
Waller et al. (2019) [25]	2	2	2	6
Woodhall et al. (2007) [39]	2	2	2	6

* Total score.

## Data Availability

Data are available upon reasonable request from the corresponding author.

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
