# Peer review of "Vaccine Hesitancy among European Parents—Psychological and Social Factors Influencing the Decision to Vaccinate against HPV: A Systematic Review and Meta-Analysis"

_vaccines, 2024, doi:10.3390/vaccines12020127_

Round 1
Reviewer 1 Report
Comments and Suggestions for Authors
I think the entire paper is very well written. One point I would like to point out is that the fixed effects model and the random effects model are used differently in the meta-analysis. What criteria are used to determine which one to use? Classically, fixed-effects models have been used, but there is a growing trend toward using random-effects models, which express the results in terms of distributions, rather than fixed-effects models, which aggregate the results of multiple studies with different study designs into a single result. The criteria for this distinction in use should be clearly stated in the methodology.
Author Response
Thank you very much for the time you took to review our manuscript. Your valuable feedback is very much appreciated. The manuscript was also sent over for a torough review by a specialized native english speaker, and appropriate modifications have been made throughout the text. Changes are highlighted in red. As per the Cochrane Handbook of Systematic Reviews (https://training.cochrane.org/handbook/current/chapter-10), “the random-effects meta-analysis approach incorporates an assumption that the different studies are estimating different, yet related, intervention effects (DerSimonian and Laird 1986, Borenstein et al. 2010). The approach allows us to address heterogeneity that cannot readily be explained by other factors. A random-effects meta-analysis model involves an assumption that the effects being estimated in the different studies follow some distribution. The model represents our lack of knowledge about why real, or apparent, intervention effects differ, by considering the differences as if they were random. In a heterogeneous set of studies, a random-effects meta-analysis will award relatively more weight to smaller studies than such studies would receive in a fixed-effect meta-analysis. This is because small studies are more informative for learning about the distribution of effects across studies than for learning about an assumed common intervention effect.” This is not a problem in the present study because no Randomized Trials were eligible for inclusion. “The random-effects method and the fixed-effect method will give identical results when there is no heterogeneity among the studies. When heterogeneity is present, a confidence interval around the random-effects summary estimate is wider than a confidence interval around a fixed-effect summary estimate.” This will happen whenever the I2 statistic is greater than zero. Given all the recommendations above, whenever low or no heterogeneity was outlined, a classic Fixed-Effect model was applied, providing similar, if not identical results to a Random-effects model. However, when high heterogeneity was discovered, the decision was undertaken to use the Random-Effects model, as it could provide more accurate results. This was not performed to ignore heterogeneity (discussed in the discussion settings and further explored by the sensitivity analyses and as part of the bias assessment). Given that no significant explanations were found for the increased heterogeneity in some analyses, the random effects were used, given that "the assumption implies that the observed differences among study results are due to a combination of the play of chance and some genuine variation in the intervention effects." Where no or low heterogeneity was observed, a Random-Effects model would have provided similar to identical results. There is a continuous debate regarding the choice of one versus the other, as mentioned in the suggestions, and definitely, the decision was not based solely on the test for heterogeneity but also on the differences between smaller and larger studies and possible causes of bias, as per the recommendations above. Therefore, the reviewer should explicitly state whether they want us to apply the random effects in all scenarios and perhaps provide the alternative OR values in the Supplementary Material (or even the main text). We are willing to add the necessary changes, should the reviewer wishes us to. Additional explanations were provided in the Material and Methods section, subsection 2.7 highlighted in red.
Reviewer 2 Report
Comments and Suggestions for Authors
Thank you for sharing your article on HPV vaccine hesitancy among parents. The following comments may help to improve the article.
L36-37: Please be more specific what you mean by high performance tests in contrast to cytology (L40). Same in L 41, what exactly is meant by "using HPV testing"?
L36: Is it age in years? Please revise.
L37: Is it age in years? Is it 35 and 45 years or 35-45 years; please clarify.
L47: Do the 107 countries refer to the WHO member states? For readers not in the field, please be more specific concerning the term "gender-neutral programs".
L54: It is not quite clear what you mean by the recommended age (9-15 years) varies significantly between countries. Do you mean the age of vaccination that, however, should be stated through the license of the vaccine? Or do you mean the recommended age related to the reimbursement of vaccine costs?
L62: Why do you restrict your article on the decision making of parents? The age group targeted, i.e., 9-15 years of age, could/should be involved in the decision making too. Also, please be more specific regarding the term "parents" as caretakers or legal adult representatives could also be in charge of decision making among the targeted age group. Looking at Table 1, you seem to have indeed included single parents, caretakers and accompanying adults. In line with this, please reconsider and revise your search strategy as well as you eligibility criteria.
General comment: Throughout your article, please pay attention when reporting age and add the correct unit, e.g., years.
L89: Please clarify if you applied any restrictions concerning e.g. language or article type.
L92: Again, why did you limit your research to parents? Also, someone at the age of 9-15 years can at least be involved in the decision making as stated above.
L102: Did articles have to have an abstract available?
L107/118: Please be more specific concerning the "relevant information" stated.
L118-124: It is not fully clear which variables listed refer to parents and which ones to their children; please revise.
L117: Please clarify if parents had to have more than one child.
L120: Family history of any cancer or restricted to cancer onsets related to the topic of your manuscript?
L122: Regarding vaccine costs, did you also look into implementation strategies covering vaccine costs as stated in the introduction of your manuscript?
Figure 1: Please be more specific concerning the registers you seem to have used to find articles meeting your criteria. If you have screened 4206 articles in total, how did you screen the 4206-4073=133 articles that were stated not to be screened manually or automatically?
Table 1: Looking at the age of children stated, I am not sure if subjects at the age of up to 19 years (David, 2022; Haesebaaert, 2014; Mari, 2022, Rousset-Jablonski, 2021) are still considered as children. Also, should articles be included in the age of children does not seem to be stated (Donders, 2007; Feiring, 2015; Schreiber, 2015; Sypien, 2022)?
L164: Please check the journal's guidelines for the number of decimal places to be included when reporting p-values; please report figures in a consistent manner throughout your article, e.g. p= .69 (L185) or p=0.04 (L192).
Comments on the Quality of English LanguagePlease see above.
Author Response
Thank you very much for your thorough review of our manuscript. Your valuable feedback is appreciated. The manuscript was also sent over for a torough review by a specialized native english speaker, and appropriate modifications have been made throughout the text. Changes are highlighted in red. Please find our responses below.
L36-37: We added the requested detail, as provided by WHO recommendations. L41: Laboratory diagnosis of HPV infection depends on molecular techniques such as DNA hybridisation or nucleic acid amplification. Several polymerase chain reaction (PCR) methods have been developed and are readily available, and usually, when referring to HPV testing, we refer to a PCR type of test. Details were added. Changes are highlighted in red.
L36: Yes, it is in years. Details were added
L37: Same here; it refers to years, and details were added. Please excuse the confusion regarding 35 and 45 years. It means that the tests should be performed again by the age of 45 years. Changes were applied accordingly.
L47: Yes, the statement applies to member countries; changes were made to the sentence, now highlighted in red. Gender-neutral refers to vaccinating both "boys" and "girls"; a supplementary explanation is attached here. Thank you for pointing this out.
L54: We added some explanations to the statement. Indeed, the study cited here refers to the recommended age in terms of national health policies, reimbursement of costs and availability through the public health system. Details are now highlighted in red. Thank you for acknowledging this.
L62: We do not restrict our investigation to parents and have successfully included studies evaluating caretakers' decisions (parents, single parents, legal guardians etc.). This is a semantic omission as we failed to state this explicitly in the referred section of the manuscript. Thank you for pointing this out; the details have now been added. Yes, the specified age group could/should be involved in the decision-making process regarding their vaccination. However, this is neither the subject of the present investigation nor their final decision. To the best of our knowledge, in the majority of European countries, for this age group, the final decision to vaccinate children belongs to their legal guardians, who will decide whether or not to take into consideration the desires of their children. It is not the subject of the present investigation to decide whether this is appropriate, nor did any of the articles provide explicit statements regarding the preferences of the children involved. Should that have been the case and the children's wishes be a factor evaluated in individual studies influencing their parents, it would have been evaluated here, but that was not the case. Indeed, this is an excellent observation, and perhaps further studies could actually evaluate children's preferences regarding their HPV vaccination and the impact on their vaccination rates. Moreover, as pointed out, column 3 of Table 1 explicitly states that the caregivers interviewed in individual studies are not just parents. Please be aware that our search was not based solely on parents and included terms referring to legal guardians also (please see the explicit search formula in the 2.2 subsections of the Materials and Methods section. Statements such as "Twenty-five primary publications were included in the systematic review, totalling 385460 responders (parents or caregivers)" and column 3 of Table 1 clearly indicate that our methodology and results are not based solely on parents. The materials and Methods and the results sections are now extensively revised to include the term caretakers where it was not previously mentioned or where just the word parents was used. We do not consider the search strategy to need any revision as it already included these neutral terms. Please be aware that the search strategy cannot be revised as quickly as it would mean completely altering the search in databases (which already retrieved more than 9000 studies) and therefore require us to completely re-make the investigation (filtering, including and analysing publications) on the newly retrieved studies and therefore rendering a new manuscript...the reviewer should explicitly state this, should they consider it necessary, but we firmly believe that the used search strategy was very inclusive, backed up both by the impressive number of results (more than 9000 publications) and the included studies (25 publications) and their total population number (385460 parents / legal guardians/caretakers).
General comment: Please excuse this oversight; changes were made throughout the manuscript and are highlighted in red.
L89: This information was already provided. We kindly ask the reviewer to refer to subsection 2.3 Eligibility Criteria of the Materials and Methods section, where all the requested details can be found.
L92: Please be aware that the search was not limited to parents! This is just a semantic oversight from the authors, who failed to mention appropriate synonyms whenever necessary, but the search strategy included neutral terms. Synonyms are now added here, too. The explanation regarding the age group 9-15 is given above, and we firmly believe the children's opinion on their vaccination is valuable; however, it does not represent the subject of the present investigation.
L102: No. We did not exclude the articles with no available abstract, only those with no available full text. Please be aware that when exporting publication data from the databases in the Excel file, not all articles have an available abstract, but that does not mean that the full-text article cannot be consulted. Please refer to our Supplementary material in the Excel spreadsheet for the detailed workflow where these details can be followed. Exclusion criteria already explicitly specify what articles were excluded.
L107/118: We kindly ask the reviewer to refer to subsection 2.7, Synthesis of Methods and Results, where the requested details are already provided. Every other mention of "relevant information" refers to this explanation. We believe there is no need to duplicate information through the manuscript as it can differ the reader from the essence (such as the referred subsection 2.5 data collection process, which emphasizes the number of experts involved in collecting relevant information rather than explicitly outlining all the information sought after, which can always be found in the subsection above.) We are willing to duplicate this if the reviewer considers it necessary. Please clarify.
L118-124: Thank you for pointing this out. As explained above, this article focuses on parents or caretakers and the particularities or factors that could influence their decision to vaccinate the children in their care. Unless explicitly stated (e.g. number of children or child's gender), all variables refer to caretakers.
L117: There were no restrictions applied regarding the number of children. A short explanation is now added.
L120: The variable family history of cancer does not refer to cervical cancer, nor does it imply a specific localization. Nevertheless, we understand that assumptions could be made; therefore, the word any was explicitly added.
L122: Yes. These details were investigated, and results can be consulted in the HPV vaccine cost subsection of the Results section. The heterogeneity of data presented in individual studies did not allow for a meta-analysis to be undertaken; thus, results are presented plainly in the form of a systematic review. None of the included articles actually provided such a variable (strategies regarding reimbursement or implementation strategies) influencing caretakers with respect to the number of guardians intending / already vaccinating their children. All results were presented as factors influencing decision but did not provide direct comparisons, making calculating the effect of measure impossible. Hence, results are presented like this.
Figure 1: The authors are unsure what information the reviewer wants here. All 9322 retrieved articles are laid out as to the registry they belong to (3215 from EMBASE, etc.). The detailed workflow can be consulted in the Supplementary Material, where the reviewer can follow, study by study, the filtering and inclusion process, clearly laid out on different spreadsheets. Regarding the reports sought for retrieval and then assessed for eligibility, this can only be performed manually...this means filtering the full-text articles through inclusion and exclusion criteria and cannot be performed automatically. This PRISMA flow chart (Figure 1) is actually available online in a ready-to-fill form and is provided as-is. Please let us know if you wish us to alter the form to explain how reports are assessed for eligibility MANUALLY.
Table 1: Although the children's age was not explicitly stated in the article, it is clear that the interviewed guardians were in charge of the decision to vaccinate children in their care, and therefore, the assumption was made that the children were not of legal age to decide for themselves, and therefore could be classified as children and not adults. Although this is debatable, the authors believe this is not a definitive factor to justify the respective publications' exclusion (as the interviewed persons were still responsible for the decision-making process). Instead, they can be reflected in the bias assessment.
L164: The authors did not identify specific guidelines. The editor can kindly provide some input regarding the preferences here. Otherwise, adjustments were made throughout the manuscript for consistency. Thank you for pointing this out.
Reviewer 3 Report
Comments and Suggestions for Authors
Thank you for the manuscript.
Your research is mainly confined to the affluent nations.
There are no reports included from 3rd world nations. Perhaps the search terms you chose did not include any reports from 3rd world nations.
Are the reasons provided by you for vaccine hesitency any different from those reports from 3rd world nations from Asia, Africa and South America?
It would be prudent to comment that HPV is a gender neutral vaccine: it causes cancer in both women and men.
Did you explore if the hesitency was for vaccinating only girls or for boys as well?
Immigrant populations are socially, financially, culturally, religion-wise distinct from the mainstream populations. They are often not insured for health needs and are underprivileged. Hence, it is not surprising that these factors stood out in your review.
It is also pertinent to note that heistancy of such populations towards vaccinatin is not towards HPV only, but extends to vaccination in general.
Author Response
Thank you very much for the time you took reviewing our manuscript. The manuscript was also sent over for a torough review by a specialized native english speaker, and appropriate modifications have been made throughout the text. Changes are highlighted in red.
Yes, the research is confined to the European region, as this was the intended purpose of the present investigation. This is also stated in the title and in various sections of the article and, of course, was stated as inclusion criteria. Even so, more than 50 studies (as shown in Figure 1 and in the Supplementary Material) were undertaken in regions other than Europe. Although this is of utmost importance and should be investigated, the vaccine hesitancy in other regions was not the purpose of this manuscript; instead, it deserves a separate investigation. Thank you very much for this compelling and valuable observation. On further investigation, some factors can be considered as superposable in all regions, while the 3rd world nations also report some worrisome statistics regarding vaccinations and the availability of such products. Some notes were added in the discussion settings with appropriate citations. Thank you for the insight!
Our search formula was gender-neutral (including terms such as "child*"). Results clearly show this, and an overview can be consulted in Table 1, column 5 regarding the gender of children (trans-gender, female, male, both or not specified in individual studies). Hesitancy was explored for both females and males (regarding the child's gender), and results can be consulted in the subsection child's gender (Figure 13).
Yes, results in immigrant populations are unremarkable and were discussed in the discussion setting, along with recommendations. Thank you for also pointing out that this hesitancy extends beyond just the HPV vaccination, which is a logical inference. However, we did not evaluate vaccination rates regarding different vaccines in such populations. We did, however, manage to correlate previous misvaccinations (regardless of the immigrant status of caretakers) with HPV vaccination rates, and results can be consulted in the manuscript and are further discussed in the discussion settings.
Thank you for your kind and insightful observations.
Reviewer 4 Report
Comments and Suggestions for Authors
The report delves into the factors contributing to the reluctance of European parents to vaccinate their children against the Human Papillomavirus (HPV), given the suboptimal adherence to vaccination programs in the region. Conducted through a comprehensive review of four databases as of September 2023, the study identified 25 publications involving 385,460 respondents. Noteworthy findings include the positive impact of variables such as immigrant and employment status, religious beliefs, parental age, and the child's gender on the decision to pursue vaccination. Furthermore, the study reveals that previous vaccine experience, perceived safety and efficacy, and maternal abnormal cervical cytology screening (CCS) results significantly influence vaccination decisions. Conversely, factors such as caregivers' education, gender, civil status, number of children, family history of neoplasia, and mother's CCS attendance did not exhibit a substantial impact on vaccine hesitancy. The research underscores the presence of various demographic, social, economic, and psychological barriers contributing to parental hesitancy, underscoring the necessity for targeted information, educational initiatives, and vaccination campaigns tailored to specific at-risk demographics. While commending the well-crafted study, I offer two suggestions. Firstly, for the presentation of Newcastle Ottawa Scale data in Table 2, employing a heatmap would enhance visualization and comprehension. Secondly, it's worth noting that prior studies have shed light on the challenges in implementing HPV vaccination, particularly in low- and middle-income countries, as evidenced by the following papers with the DOIs: 10.1007/s40273-016-0451-7 and 10.1111/ajco.13513
Author Response
Thank you very much for the time you took reviewing our manuscript. Your suggestions are valuable. The manuscript was also sent over for a torough review by a specialized native english speaker, and appropriate modifications have been made throughout the text. Changes are highlighted in red. In supplementary material 2, we added a graphic representation of the NOS scoring to enhance comprehension as suggested. Thank you for suggesting these relevant publications. They are now properly cited within the text.
Round 2
Reviewer 2 Report
Comments and Suggestions for Authors
Thank you for sharing your revised manuscript that reads much better. My comments were addressed sufficiently.
Comments on the Quality of English LanguagePlease see above.